# Rethinking the One-shot Object Detection: Cross-Domain Object Search

## ABSTRACT

One-shot object detection (OSOD) uses a query patch to identify the same category of object in a target image. As the OSOD setting, the target images are required to contain the object category of the query patch, and the image styles (domains) of the query patch and target images are always similar. However, in practical application, the above requirements are not commonly satisfied. Therefore, we propose a new problem namely Cross-Domain Object Search (CDOS), where the object categories of the query patch and target image are decoupled, and the image styles between them may also be significantly different. For this problem, we develop a new method, which incorporates both foreground-background contrastive learning heads and a domain-generalized feature augmentation technique. This makes our method effectively handle the object category gap and domain distribution gap, between the query patch and target image in the training and testing datasets. We further build a new benchmark for the proposed CDOS problem, on which our method shows significant performance improvements over the comparison methods.

## CCS CONCEPTS

• Computing methodologies → Artificial intelligence; Computer vision; Object detection;

## KEYWORDS

cross-domian, object search, one-shot object detection, domain geneteal-ization

**ACM Reference Format:**

. 2024. Rethinking the One-shot Object Detection: Cross-Domain Object Search. In *Proceedings of Make sure to enter the correct conference title from your rights confirmation emai (Conference acronym 'XX).* ACM, New York, NY, USA, 9 pages. https://doi.org/XXXXXXX.XXXXXXX

## 1 INTRODUCTION

Object detection is one of the fundamental problems in computer vision. The conventional approach relies on training with a large volume of annotated data and then testing on specified categories, constituting a closed-classes task that is difficult to extend to novel categories. In contrast, One-shot Object Detection (OSOD) tasks [2, 12, 32, 36] aim to break the limitation of this closed-classes setting

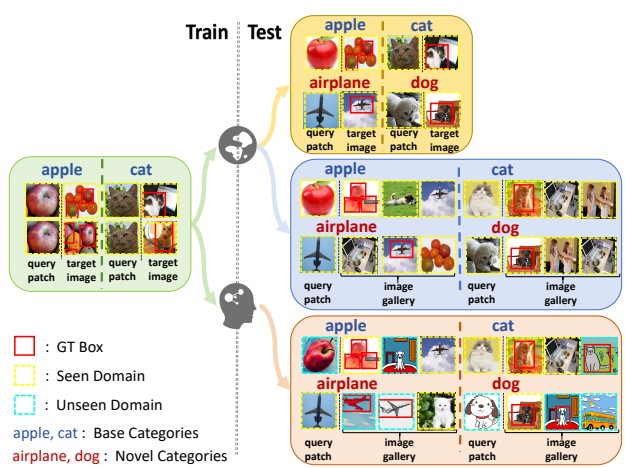

**Figure 1: Cross-Domain Object Search. During training, we possess images of real-life scenes featuring multiple base categories, such as apple and cat. During testing, we aim to search for novel categories, like airplane and dog, within unseen domains, such as watercolor and oil.**

and make object detection into an open-classes task. The inspiration for OSOD comes from humans' remarkable ability to recognize and locate objects by providing them with an example object that they have never seen before. In other words, humans can rapidly learn new concepts and features from a single example and then identify the same category of objects in other scenes.

As the setting of the OSOD task, given a query patch, and a target image containing objects of the same category as the query patch, the goal is to identify the objects (in the target image) of that category. We have a question about this setting: Does the target image necessarily contain objects of the same category as the query patch? In real-world applications, a common scene may be identifying the objects of interest from a image gallery (no matter whether containing the query patch), but not manually selecting the images with the desired detection objects as input. Further, it is also important that the image gallery should be broad, potentially containing data from various image styles (domains).

From the above two perspectives, in this work, we propose to study a new problem, *i.e.*, Cross-Domain Object Search (CDOS). Different from the classical OSOD, we do not require all the target images in the gallery to contain the object category of the query patch. We also allow the target images to have various styles, which can be different from the query patch, and even have not been seen in the training dataset, as depicted in Figure 1.

The proposed CDOS is a very challenging problem. The first challenge comes from the object (category) similarity measurement. In OSOD, the main problem is to find the most similar object to the query patch in a target image. In CDOS, we have to first judge

whether the target contains the desired object. The second challenge comes from the domain gap. In our problem, we allow the target images within various domains, *e.g.*, a cartoon-style image, which has the domain gap with the query patch. Moreover, following the domain generalization problem [30], we propose to use all the natural images as the training set, while the style images during testing are not seen during training.

To address these two challenges, we present a new approach for CDOS. Specifically, we first propose a new module for foreground-background contrastive learning. This module is composed of two parts. The foreground contrastive learning head aims at improving the network's ability to classify foreground objects, thereby reducing category confusion. The background contrastive learning head is designed to bolster the network's capacity to distinguish background, thus mitigating background misjudgment (false positives). Besides, to address the challenge of open-domain issues, we implement a random feature perturbation augmentation on the feature extraction network during training. This data augmentation strategy diversifies the training scenarios by altering the statistical properties of images, which enhances the model's robustness when encountering data from unknown domains. In summary, the main contributions of this work are:

- We extend the one-shot object detection task and propose a new and practical problem, *i.e.*, Cross-Domain Object Search (CDOS), which enables a convenient search of objects of interest in the category-free and domain-various image galleries.
- We develop the first baseline method for CDOS, which constructs an abundant number of foreground and background samples for contrastive learning, to enhance the model's feature discrimination capabilities. Furthermore, a feature augmenter module is applied to narrow the domain gap between the query and the target images, also the training and testing data.
- We build a new large-scale image dataset namely Multi-Style Object Search Benchmark (MSOSB), which provides a benchmark to facilitate the training and testing of CDOS problems. Extensive experiments on it demonstrate the effectiveness of our method, which significantly outperforms the state-of-the-art methods. The MSOSB is made publicly available at https://2899253375.github.io/blog/.

## 2 RELATED WORK
### 2.1 One-shot Object Detection

In the domain of computer vision, One-shot Object Detection (OSOD) presents a unique object detection task. The objective is to detect objects within an image that are of the same class as a presented query patch after having only a single instance of the object to reference, without further fine-tuning for new categories. This differs from conventional object detection methods, such as Faster-RCNN [24], which generally require extensive labeled data to train models for recognizing specific object classes. Works like [2, 12, 32, 36] have made notable contributions to OSOD by employing feature fusion to obtain directly transferable meta-knowledge, thereby generalizing this meta-knowledge to new categories. Among these,

BHRL [32] introduced an innovative Instance Hierarchical Relation (IHR) module that achieves superior results. In machine learning, such capabilities are particularly critical for scenarios where collecting or annotating vast amounts of data is challenging. The OSOD task is akin to our proposed object search task, but OSOD evaluations focus on paired query patches and target images and do not consider searching for objects within an image gallery, leading to a high rate of false positives in images without the objects of interest. We have developed a novel method for the object search task that mitigates these issues.

### 2.2 Single Domain Generalization

In this study, we delve into Domain Generalization (DG), which aims to learn a robust model from multiple source domains that can be effectively generalized to any unseen target domain. Single Domain Generalization (SDG) [30] represents the more challenging extreme of DG, where only one source domain dataset is used for training with the goal of adapting to multiple unseen target domains. To address this challenge, various data augmentation algorithms have been designed to enhance the diversity of training data, as proposed in [19, 23, 30]. A domain augmentation module for synthesizing images was introduced in [30], while [19] incorporated synthetic feature statistics to simulate the uncertainty of domain shift during training. To standardize SDG training, [5] incorporated various visual impairments as augmentations and devised a novel attention consistency loss. A new image meta-convolutional network capturing more domain-generalizable features was developed in [29]. There are also approaches specific to SDG for Object Detection. For instance, [31] proposed a cycle-consistent disentangled self-distillation method that disentangles from domain-specific representations without domain-related annotations (e.g., domain labels), and [28] employed pre-trained visual-language models to introduce semantic domain concepts through text prompts. In the context of CDOS, similar challenges to SDG are evident, such as training with real-world images and, during testing, the query patch and target image may appear in any domain. Particularly challenging is when the styles of the query patch and target image differ, for which we have applied feature augmentation in the twin branches of the Siamese network [3] to implement feature random augmentation.

### 2.3 Out-of-distribution Detection

Most existing machine learning models are trained based on the closed-world assumption, where it is assumed that the test data will share the same distribution as the training data, known as In-Distribution (ID). However, when models are deployed in open-world scenarios [1], the test samples can be Out-Of-Distribution (OOD), significantly increasing the difficulty of recognition. Distribution shifts can be caused by semantic shifts (e.g., samples from different categories) or covariate shifts (e.g., samples from different domains) [33]. Generalizing models to OOD data is a natural capability of humans that is challenging to replicate in machines. In the context of CDOS, both query patches and target images may originate from out-of-domain data (novel categories and unseen domains). Overall, the CDOS task can also be considered a form of Out-of-distribution Detection, capable of searching for objects in novel categories as well as unseen domains.

## 3 MSOSB: A MULTI-STYLE OBJECT SEARCH BENCHMARK

Datasets containing a variety of style data can significantly enhance the evaluation of a model's domain generalization capabilities. Thus, for CDOS methods, conducting assessments on datasets with a wide range of image styles is essential. As shown in Table 1, existing multi-style object detection datasets can be categorized as follows: (1) General objects: Clipart [14], Watercolor [14], Comic [14]; (2) Traffic scenes: KITTI [9], Cityscapes [4], BDD100K [35], FoggyCityscapes [26], UFDD [22], RTTS [17], Sim10K [15]; (3) Face detection: WIDER FACE [34]. The datasets Clipart [14], Watercolor [14], and Comic [14] include abstract, artistic, and comic images, respectively. Understanding these types of abstract imagery allows for direct investigation into how models infer high-level semantic information. The Clipart [14] comprises approximately 1K images across 20 categories, aligned with Pascal VOC [7]. The Watercolor and Comic datasets contain around 1K training images and 1K testing images across six categories. **Overall, the existing style datasets for common objects are modest in size and offer a limited array of categories, falling short of the needs for assessing domain generalization models in generic object detection tasks. Consequently, we have developed MSOSB: a multi-style object search benchmark.** We will detail MSOSB's key aspects in the following sections.

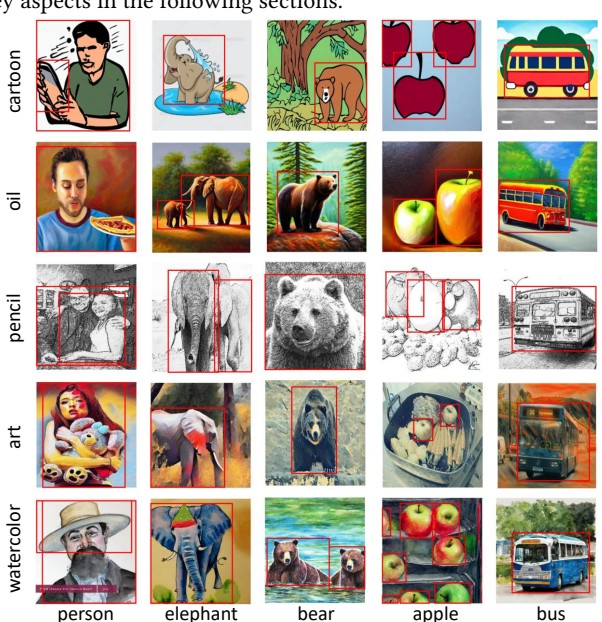

**Figure 2: Collected data from five styles. Images typically contain not only the objects and complex backgrounds but also objects of various other categories.**

### 3.1 Benchmark Construction

We aim to construct a multi-style benchmark for the CDOS task, sharing categories with the MS COCO [21]. This benchmark will encompass styles such as cartoon, oil, watercolor, pencil, and art. To curate images in these styles, we initially selected suitable images for object detection tasks from the image style classification

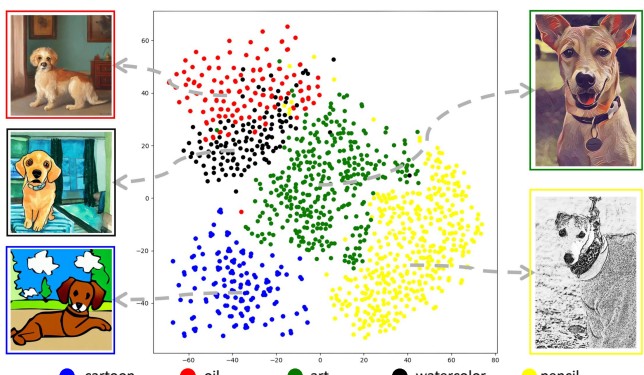

**Figure 3: The t-SNE of feature-level style statistics, $\Phi = [\mu, \sigma]$, derived from the outputs of ResNet 50 for samples of the dog category. Samples are clustered based on domain features.**

datasets PACS [18] and OfficeHome [27]. Subsequently, we scoured the internet for images in these styles, aiming to represent each category adequately. Given the impressive capabilities of current generative models and their proliferating presence online, bringing generated images ever closer to our daily lives, we utilized the stable diffusion model [25] to generate images for each category and style, followed by a meticulous selection process. The images were then annotated and verified meticulously, following the annotation scheme of MS COCO [21], ensuring the benchmark's versatility and reliability. For the Clipart [14], Watercolor [14], and Comic [14] datasets, we conducted a selection and reorganization process before integrating them into our dataset. Through t-SNE analysis, we observed that the distributions of Clipart [14] and Comic [14] were nearly identical, leading us to merge these datasets into the cartoon style. To further diversify the dataset's style, we applied BaiDu's style conversion API to the validation set of MS COCO [21] and all data from Pascal VOC [7], generating pencil and art-styled data.

In summary, as depicted in Figure 2 and the t-SNE visualization distribution in Figure 3, we have constructed a benchmark comprising 80 categories. This benchmark includes 999 images from Clipart [14], 1905 images from Watercolor [14], 1905 images from Comic [14], 264 images from PACS [18], 170 images from OfficeHome [27], 1004 images sourced from the internet, 37624 images generated by stable diffusion [25], and 32275 images produced via BaiDu's style conversion API, culminating in a total of 75146 images in the MSOSB.

### 3.2 Data Cleaning and Annotation

For the existing datasets Clipart [14], Watercolor [14], Comic [14], PACS [18], and OfficeHome [27], we assigned three researchers to screen and adjust the annotations for the Clipart [14], Watercolor [14], and Comic [14] images, with the principle that the images must be readable and clear, and any misaligned annotations corrected. Three researchers were tasked with selecting images from the PACS [18] and OfficeHome [27] classification datasets, as well as collecting high-quality images of similar styles from the internet. The selection criteria were that the images should belong to the cartoon, oil, or watercolor styles, and be suitable for object

**Table 1: Relative to Other Styled Object Detection Datasets.**

| Name | Year | Images | Classes | Styles | Application Scenes | Image source |
|------|------|--------|---------|--------|-------------------|--------------|
| Pascal VOC [7] | 2010 | 21,493 | 20 | 1 | General objects | Internet |
| KITTI [9] | 2013 | 14,999 | 1 | 1 | Traffic scenes | Shooting |
| MS COCO [21] | 2014 | 123,287 | 80 | 1 | General objects | Internet |
| Cityscapes [4] | 2016 | 3,475 | 8 | 1 | Traffic scenes | Shooting |
| Sim10K [15] | 2016 | 10,000 | 1 | 1 | Traffic scenes | Game engine |
| WIDER FACE [34] | 2016 | 32,000 | 1 | 1 | Face detection | Shooting |
| FoggyCityscapes [26] | 2018 | 3,475 | 8 | 1 | Traffic scenes | Shooting |
| Clipart [14] | 2018 | 1,000 | 20 | 1 | General objects | Internet |
| Watercolor [14] | 2018 | 1,905 | 6 | 1 | General objects | Internet |
| Comic [14] | 2018 | 1,905 | 6 | 1 | General objects | Internet |
| UFDD [22] | 2018 | 884 | 1 | 1 | Traffic scenes | Shooting |
| RTTS [17] | 2018 | 9,109 | 5 | 1 | Traffic scenes | Various |
| BDD100K [35] | 2020 | 41,986 | 10 | 12 | Traffic scenes | Shooting |
| **MSOSB** | **2024** | **76,146** | **80** | **5** | **General objects** | **Various** |

detection tasks (the target objects' area should not be too large), followed by format conversion, meticulous selection, and annotation. Subsequently, we deployed ten researchers to generate cartoon, oil, and watercolor style images using stable Diffusion [25] via Google's Colab, with categories matching those of MS COCO [21]. For each style and category, 500 images were generated with varying sizes, ranging from 512 to 2048 pixels. Given the variable quality of some generated images, a rigorous selection and annotation process was necessary. Since the MS COCO [21] validation set and all Pascal VOC [7] data are already annotated, we utilized BaiDu's style conversion API for stylization, randomly generating pencil and art-style images. It's noteworthy that this stylization does not alter the content of the images, hence the original annotations can still be used.

Finally, we standardized the annotation files into Pascal VOC [7] and MS COCO [21] formats. The first two digits of the image names indicate the style, for example, "01" for cartoon, followed by a sequence number representing the image's order within its style category, such as "01000000002" for the second image in cartoon style.

### 3.3 Dataset Characteristics

The MSOSB introduces a series of challenging settings that rigorously test the limits of CDOS models, with challenges arising from:

**Variability in object size and aspect ratio.** The MSOSB dataset includes objects of different sizes and shapes. This requires models to have strong abilities to recognize and distinguish objects in various visual situations.

**Complex Backgrounds and Overlapping Objects.** Images within the MSOSB dataset typically exhibit complex backgrounds and overlapping objects, reflecting the conditions of cluttered environments around the objects in the real world.

**Multi-category multi-object scenes.** Each image in the MSOSB may contain multiple objects with different categories to measure the models' ability of robust object classification/localization.

**Cross-Domain Variability.** Since MSOSB shares categories with MS COCO [21], it can be combined with MS COCO [21] to use together in six styles, further challenging the domain generalization capabilities of CDOS models.

These challenging settings are meticulously planned to simulate complex real-world scenes, thereby providing a comprehensive benchmark for advancing CDOS techniques.

## 4 PROPOSED METHOD

In this section, we first define the setup for the CDOS problem and then elaborate on how the key components of our proposed method (CrDoOS) address the CDOS task.

### 4.1 CDOS Problem Formulation

Similar to OSOD, object catogories are divided into base catogories $B$ and novel catogories $N$, where $B \cap N = \emptyset$. The domains of the data are split into seen domains $S$ and unseen domains $U$, with $S \cap U = \emptyset$. Given an arbitrary query patch, CDOS aims to detect targets within an image gallery, which contains a large and complex assortment of unknown images, that match the category of the query patch. Like OSOD, the CDOS task is trained using data from the base catogories and seen domains. After training, it can generalize to directly search for objects in an image gallery using just a single query patch, where the image gallery includes images from both base and novel catogories, as well as images from various domains. Unlike the OSOD task, the images in image gallery does not necessarily contain objects of the same category as the query patch in CDOS, and the styles of the target image and query patch may not match. This significantly increases the difficulty and aligns more closely with real-world scenarios.

### 4.2 Framework

We introduce a CDOS framework inspired by BHRL [32], with its overall architecture depicted in Figure 4. This structure utilizes a two-stage object detection paradigm and employs a Siamese network [3] to extract features from both the query patch and the target image. Subsequently, a feature fusion module aggregates the features of the query patch and proposals of target image for classification and regression tasks. Building on this foundation, we have incorporated a foreground-background contrastive learning heads, enabling the network to fully leverage the features of objects across various categories and their background for contrastive

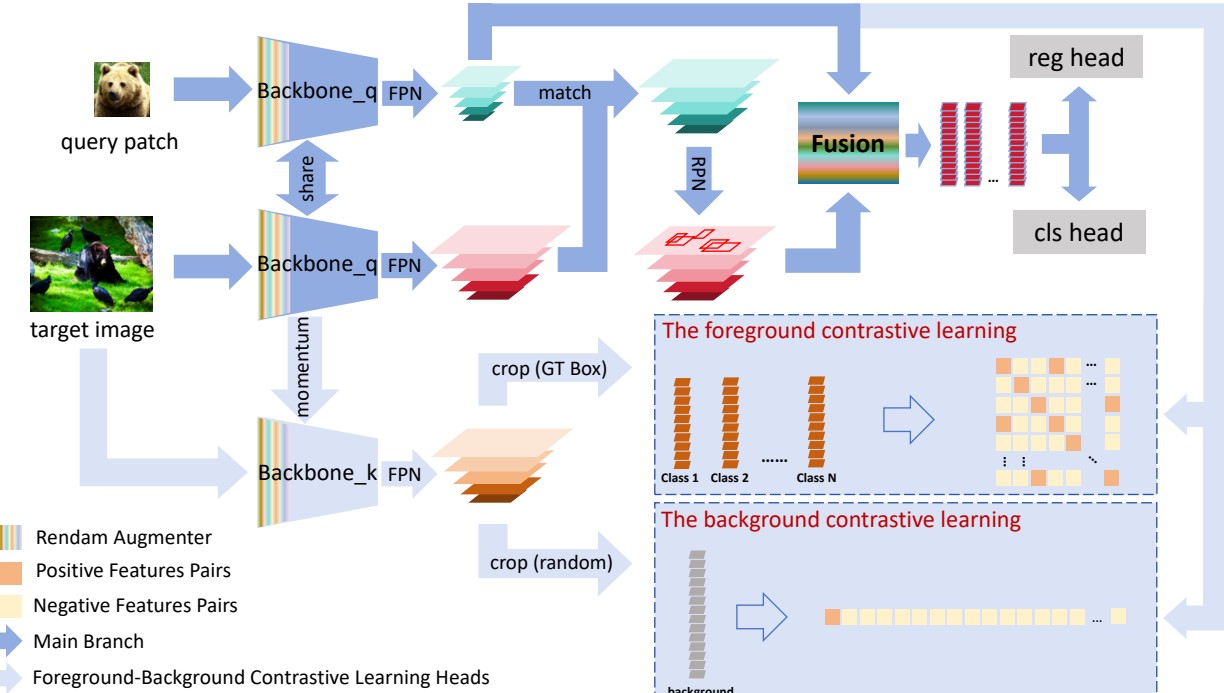

**Figure 4: Overview of CrDoOS. The diagram illustrates the comprehensive training workflow of our method, where both Backbone_q and Backbone_k are ResNet 50. The parameters of Backbone_k are momentum-updated based on Backbone_q. The light blue sections represent our proposed Foreground-Background Contrastive Learning heads. The training process involves both the deep and light blue sections, using a pair of query patch (fed into the first branch of the Siamese network) and target image (fed into the second branch of the Siamese network and the Foreground-Background Contrastive Learning heads). For testing, the process traverses the deep blue section, where the query patch (fed into the first branch of the Siamese network) and the image gallery (fed into the second branch of the Siamese network) are utilized.**

learning. This addition enhances the network's discriminative capability regarding features without increasing inference parameters and computation time. To address the domain discrepancy between the query patch and the target image, we have incorporated a Feature Random Augmenter into both branches of the Siamese network [3]. This augmenter mainly operates by randomly perturbing the mean and variance of shallow features within the Siamese network [3] to achieve feature augmentation, thereby exposing the network to a broader range of domains and enhancing its ability to search for objects in images of various styles. Overall, our method presents two primary contributions: firstly, we introduced a foreground-background contrastive learning heads (see Subsection 4.3), and secondly, we implemented a feature random augmenter within the Siamese network [3] (see Subsection 4.4).

### 4.3 Foreground-Background Contrastive Learning Heads

A classical method–MoCo [10], by employing momentum update mechanisms for feature comparison, has exerted a beneficial and profound impact on self-supervised learning. This approach demonstrates that utilizing a large set of features for contrastive learning enhances the backbone's capabilities in feature extraction and discrimination. Given the severe foreground confusion and background misclassification encountered when directly applying OSOD

methods to object search tasks, we introduced a momentum-updated foreground-background contrastive learning heads. This addition improves the backbone's discriminative ability, thus mitigating issues of foreground confusion and background misclassification. The heads are divided into two parts.

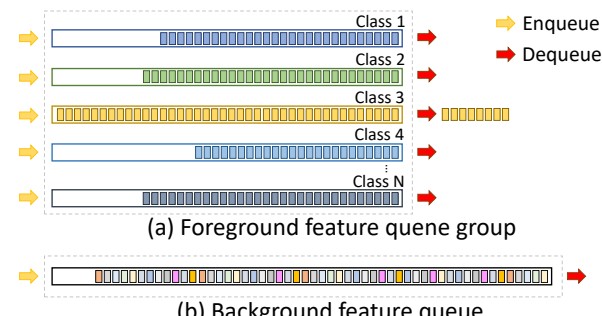

**Figure 5: Foreground feature queue group and background feature queue. (a) is the foreground feature queue group, where each category has a separate queue for storing foreground features. (b) is the background feature queue, where the network randomly crops a piece of background feature from each target image for storage.**

**Foreground Contrastive Learning.** We use a foreground feature contrastive learning strategy, which aims to resolve issues of foreground object ambiguity. Specifically, as shown in Figure 4, the features of the target images are extracted using the Backbone_k, which employs momentum-updated parameters in combination with the Feature Pyramid Network (FPN) [20], as described in following

$$\theta_{Backbone\_k_{(t)}} = m \cdot \theta_{Backbone\_k_{(t-1)}} + (1-m) \cdot \theta_{Backbone\_q_{(t)}} \quad (1)$$

where $\theta_{Backbone\_k}$ denotes the parameters of the Backbone_k, and $\theta_{Backbone\_q}$ represents the parameters of Backbone_q, $t$ denotes the number of iterations, and $m$ stands for the momentum update parameter.

For the supervision of this branch, the foreground features are cropped from the image using Ground Truth (GT) boxes of base categories and processed through a projector layer. These features are stored and updated in a categorized queue group, as illustrated in Figure 5 (a).

During this process, the features of objects from the same category as the query patch are considered positive samples, while those from different categories serve as negative samples. With the positive/negative samples, we apply the following loss function in Equation (2) [16] to train the network by contrastive learning

$$L_{\text{Foreg}} = \sum_{i \in I} L_{\text{Foreg},i} = \sum_{i \in I} -\frac{1}{|P(i)|} \sum_{p \in P(i)} \log \frac{\exp(z_i \cdot z_p/T)}{\sum_{a \in A(i)} \exp(z_i \cdot z_a/T)} \quad (2)$$

where $L_{\text{Foreg}}$ denotes the foreground contrastive learning loss, $T \in \mathbb{R}^+$ is a scalar temperature parameter, $z_p$ is a sample of the same category as $z_i$, and $z_a$ represents samples from other categories, $A(i)$ represent the set of samples excluding $i$.

Through the above strategy, extensive contrastive learning among base category features enhances the discriminative capability of the Backbone_q, thereby alleviating foreground confusion during inference. It is important to note that in our design of the foreground feature queue group, we employ queues of the same length for different categories. *This ensures that the number of features per category is consistent, thereby facilitating a more balanced learning process across the network.*

**Background Contrastive Learning.** We then consider the background contrastive learning head. Our basic idea is to build an individual discrimination task from the vast image backgrounds. *By maximizing the separation between object features and background features, we can enhance the network's ability to discriminate between foreground and background regions.*

This way, our background contrastive learning head involves contrasting positive sample pairs against a large pool of negative samples. Specifically, we randomly crop features from the background areas of the target image to populate the background feature queue (ensuring that the cropped features are not too small to maintain rich background information), as illustrated in Figure 5(b). By storing these background features in an extensive queue, we enrich the diversity of background samples, effectively leveraging the advantages of individual discrimination tasks and contrastive learning. The features of the query patch and its augmented counterpart serve as positive samples, while all features in the background feature queue serve as negative samples.

Finally, similar to the foreground head, the loss function for this background contrastive learning is detailed in the following Equation (3).

$$L_{\text{Backg}} = \sum_{i \in I} L_{\text{Backg},i} = -\sum_{i \in I} \log \frac{\exp(z_i \cdot z_{j(i)}/T)}{\sum_{b \in B(i)} \exp(z_i \cdot z_b/T)} \quad (3)$$

where $L_{\text{Backg}}$ denotes the background contrastive learning loss, $z_{j(i)}$ is the augmented sample of $z_i$, and $z_b$ represents the background sample, $B(i)$ represent the set of samples excluding $i$.

## 4.4 Feature Random Augmenter

Considering the domain gap between the query patch and the target image, and the training and testing datasets, we further develop a feature random augmenter to narrow the domain gap.

As discussed in previous work [13], the statistics of image feature channels (mean and standard deviation) are closely related to image style. Altering these channel statistics can be seen as an implicit way to change the style of the input image, as in Equation (4)

$$\text{Sty}_{(x)} = \frac{\sigma_{(y)}}{\sigma_{(x)}}(x - \mu_{(x)}) + \mu_{(y)}. \quad (4)$$

where $x$ represents the original sample and $\text{Sty}_{(x)}$ denotes the generated sample with image style transformation. Here $\mu_{(x)}$ denotes the mean of sample $x$, $\sigma_{(x)}$ represents the variance of sample $x$, $\mu_{(y)}$ denotes the mean of the augmented sample $y$, and $\sigma_{(y)}$ represents the variance of the augmented sample $y$.

Inspired by [8], we employ Normalization Perturbation (NP) to disturb the image features, as represented by Equation (5).

$$y = \frac{\sigma_{(x)}^* \left(x - \mu_{(x)}\right)}{\sigma_{(x)}} + \mu_{(x)}^*, \quad \sigma_{(x)}^* = \alpha\sigma_{(x)}, \quad \mu_{(x)}^* = \beta\mu_{(x)} \quad (5)$$

where $\{\mu_{(x)}, \sigma_{(x)}\}$ are the channel statistics, mean and standard deviation of the sample $x$, estimated on the input features. The $\{\alpha, \beta\}$ are random noises drawn from the Gaussian distribution. The equation can be simplified to

$$y = \alpha x + (\beta - \alpha)\mu_{(x)}. \quad (6)$$

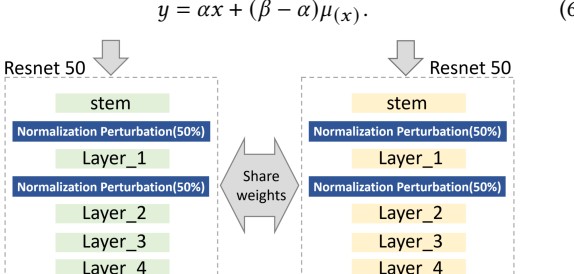

**Figure 6: Incorporate Normalization Perturbation into the Siamese Network.**

The above NP operation effectively synthesizes various potential styles by perturbing the channel statistics in the shallow CNN layers of source domain images, without altering the shape or position of the objects within the images, which is suitable for the object search tasks in this work.

We next present how to integrate the NP into our framework. Specifically, as shown in Figure 6, we use a straightforward strategy by applying the Normalization Perturbation (NP) to both branches

of the Siamese network, setting an activation probability of 0.5. It is noteworthy that the augmenters in both branches of the Siamese network operate independently, enabling the network to generate various augmentation patterns during training. This independence allows each branch to process different domains separately during training, learning domain-invariant feature representations. Based on this, the trained models are more universal across different domains and generalize well even without data from the target domain. Consequently, this aids in better matching the objects across different styles.

## 4.5 Implementation Details

We trained our model using the Stochastic Gradient Descent (SGD) optimizer for 15 epochs, employing a weight decay of 1e-4. The training was performed on 4 NVIDIA 3090 GPUs with a batch size of 16. The initial learning rate was set at 0.02 and reduced by a factor of ten after the seventh epoch. Our backbone, a ResNet 50 [11], was pretrained on a reduced version of ImageNet [6], ensuring that our model was not exposed to novel classes. The momentum value $m$ in Equation (1) was set to 0.999. For the feature queues in the foreground-background contrastive learning heads, we standardized the length of each category's foreground queue to 128, while the background feature queue was set to 65,536.

## 5 EXPERIMENTS

### 5.1 Setup

**Benchmark.** For a fair comparison, we adhere to the division method previously established in works such as [2, 12, 32], spliting the 80 categories of MS COCO [21] into four distinct combinations (split1-split4). For the testing experiments of the CDOS task, we meticulously reference the setup from image retrieval experiments, specifically choosing Split1 as our designated experimental setup.

**Metrics.** Considering the distinct setup of the CDOS task from the OSOD task, the metric for the CDOS task should calculate the search precision for each category and then compute an average across all categories. Ultimately, we utilize the AP50 metric from MS COCO [21] to evaluate the outcomes of the CDOS task. This approach ensures that the evaluation accurately reflects the model's performance across all categories, adhering to academic standards while enhancing the clarity and comprehensibility of our findings.

**Comparison methods.** As a new problem, we can not find a method to handle the CDOS problem directly. We try to include more methods for comparison. First, given the similar dataset configurations and training processes between CDOS and OSOD, we have selected several stare-of-the-art OSOD methods CoAE [12], AiT [2] and BHRL [32] as the comparison method. Additionally, in CDOS problem, a key problem comes from the domain gap from different image styles. This way, to evaluate the effectiveness of our method for pen-domain scenarios, we also include two mainstream domain augmentation modules Learning to Diversify [30] and ACVC [5] into our approach, thereby investigating the cross-domain robustness of our model.

### 5.2 Comparison with State-of-the-art Methods

As discussed above, based on OSOD, this work makes the following two extensions. **Category extension**: we expand the search space from the specific target images into the query-free retrieval gallery. **Doamin extension** we expand the image styles of the gallery from the natural image into a variety of styles. For comprehensive testing, we develop three protocols for evaluation.

**Protocol I**: **Fixed category in each domain.** We randomly selected 10 images from each style and category to serve as query patches, which were then used to search for objects within an image gallery. The gallery is built upon all the images with the same style and category of the query in the whole dataset.

**Protocol II**: **Fixed category in various domains.** We randomly selected 10 images from each style and category to serve as queries, which were then used to search for objects within an image gallery of the same category but in all styles in the dataset.

**Protocol III**: **Free category in various domains.** We randomly selected 10 images from each style and category to serve as query patches. We select the images from the dataset to form a complex gallery, without considering the style and category consistency with the query. This can be regarded as the most typical and generic CDOS setup, which is also the most challenging. Note that, without the limitation of both category and style, the gallery should have been the whole dataset, which is too large for implementation. This way, in our experiments, we randomly select 2k images as the gallery.

Table 2 display the experimental results under the three protocols. It is evident that our method has achieved an increase of 2.6% in AP50 for base categories and 1.7% in AP50 for novel categories, resulting in an overall improvement of 2.4% in the protocol I. In the protocol II, there was an enhancement of 5.3% in AP50 for base categories and 5.0% in AP50 for novel categories, leading to a total elevation of 5.2% across the test set. In the most stringent protocol III, our approach realized a significant uplift of 4.2% in AP50 for base categories and 2.3% in AP50 for novel categories, culminating in an overall boost of 3.8% across the entire test dataset. Overall, the improvements are notably substantial.

Additionally, we also follow the previous experimental setup of the classical one-shot object detection (OSOD) and conduct the experimental evaluation on MS COCO dataset [21]. The results, displayed in Table 3, show that we exclusively used the foreground-background contrastive learning heads. The data indicate that our method achieved an approximate 0.1% increase in AP50 for both base and novel categories. Although this improvement may seem modest for the OSOD task, our method showed more significant enhancements in the CDOS task, suggesting that it is particularly effective for the challenges unique to CDOS.

### 5.3 Ablation Studies

In this section, we conducted extensive ablation experiments to analyze the impact of each component of our proposed CrDoOS. We used AP50 as the primary performance metric.

**Component Analysis.** We conducted experiments based on Protocol III to verify the effectiveness of the proposed contrastive learning module and feature random augmenter, summarizing the average precision across all categories on the test dataset in Table 4. The method in the first row adopts the BHRL [32]. As shown in the second to fourth rows, the application of the contrastive learning for foreground features, background features, and the combined effect of both, resulted in improvements of 1.4%, 2.2%, and

Table 2: Comparisons of CrDoOS with state-of-the-art methods on the MSOSB in terms of AP50 under the three protocols.

| Method | Protocol I | | | Protocol II | | | Protocol III | | |
|---|---|---|---|---|---|---|---|---|---|
| | Base | Novel | All | Base | Novel | All | Base | Novel | All |
| CoAE [12] | 40.1 | 33.4 | 38.4 | 38.4 | 28.7 | 36.0 | 9.8 | 2.9 | 8.1 |
| AIT [2] | 49.7 | 37.9 | 46.8 | 46.5 | 33.9 | 43.4 | 11.6 | 3.0 | 9.5 |
| BHRL [32] | 54.8 | 39.6 | 51.0 | 52.0 | 37.8 | 48.5 | 13.5 | 3.5 | 10.9 |
| **CrDoOS** | **57.4** | **41.3** | **53.4** | **57.3** | **42.8** | **53.7** | **17.7** | **5.8** | **14.7** |

Table 3: Comparisons with state-of-the-art methods on the COCO dataset in terms of AP50 under the OSOD setting.

| Method | Base | | | | | Novel | | | | |
|---|---|---|---|---|---|---|---|---|---|---|
| | Split-1 | Split-2 | Split-3 | Split-4 | Average | Split-1 | Split-2 | Split-3 | Split-4 | Average |
| CoAE [12] | 42.2 | 40.2 | 39.9 | 41.3 | 40.9 | 23.4 | 23.6 | 20.5 | 20.4 | 22.0 |
| AIT [2] | 50.1 | 47.2 | 45.8 | 46.9 | 47.5 | 26.0 | 26.4 | 22.3 | 22.6 | 24.3 |
| BHRL [32] | **56.0** | **52.1** | 52.6 | 53.4 | 53.5 | 26.1 | **29.0** | 22.7 | 24.5 | 25.6 |
| **CrDoOS** | **56.0** | 51.9 | **52.8** | **53.5** | **53.6** | **26.3** | 28.8 | **22.9** | **24.6** | **25.7** |

2.7% AP50 on the test dataset, respectively. This benefit is attributed to the contrastive learning heads, which enhance the discriminability of the network-generated foreground and background features. As illustrated in the fifth row of the table, the feature random augmenter improved the BHRL [32] method by 1.1% AP50. This indicates that the feature random augmenter can increase the network's robustness across multi-domain datasets by introducing more style variations of training samples through perturbations of shallow features.

Table 4: Effects of each component.

| foreg contra | backg contra | Aug | $AP_{50}$ |
|---|---|---|---|
| | | | 10.9 |
| ✓ | | | 12.3 |
| | ✓ | | 13.1 |
| ✓ | ✓ | | 13.6 |
| | | ✓ | 12.2 |
| ✓ | | ✓ | 13.6 |
| | ✓ | ✓ | 14.2 |
| ✓ | ✓ | ✓ | 14.7 |

**The Impact of Different Random Augmenter Module.** We explored the integration of different Single-Source Domain Generalization methods into our method and their effects. For instance, the feature augmentation parts of Learning to Diversify [30] and ACVC [5] could be seamlessly incorporated into our network, with the final results presented in Table 5. It is evident that the methods of Learning to Diversify [30] and NP [8], which directly perturb image features, are more conducive to enhancing the model's domain generalization capability. In contrast, the fixed form of feature augmentation in ACVC [5] is slightly less effective than the first two augmentation methods. This suggests that random domain augmentation can expose the model to a wider variety of styles, thereby strengthening its domain generalization ability.

Table 5: Feature augmentation comparison results.

| Results of Various feature augmentation Methods | | | |
|---|---|---|---|
| Method | Learning to Diversify [30] | ACVC [5] | NP [8] |
| $AP_{50}$ | 14.4 | 14.1 | 14.7 |

## 5.4 Qualitative Results

In Figure 7, we visualize the test results of BHRL [32] and CrDoOS on the MSOSB. It is evident that our CrDoOS can accurately detect targets of interest categories. Compared to OSOD methods, our proposed CrDoOS generates fewer false detections.

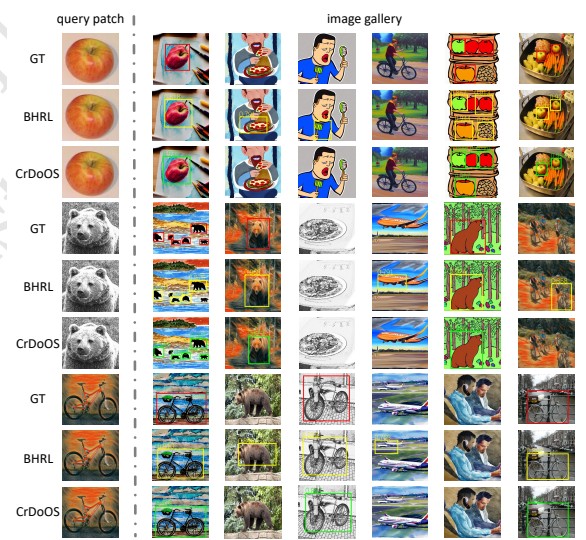

Figure 7: Visualization comparisons.

## 6 CONCLUSION

In this paper, we have proposed a novel problem of CDOS and a corresponding method, CrDoOS. Initially, we proposed a foreground-background contrastive learning heads, which have substantially improved the model's ability to discriminate features, thereby mitigating issues related to foreground confusion and background misjudgments. Then, we incorporated a feature random augmenter to strengthen the model's capability for domain generalization. Additionally, we devised the MSOSB to evaluate our CDOS method. Compared with existing OSOD methods under similar settings, our model has achieved state-of-the-art performance. We hope that our work provides practical insights and methodologies that closely reflect real-world scenarios and contributes to the advancement of open-category and open-domain challenges in the field.

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

Received 20 February 2007; revised 12 March 2009; accepted 5 June 2009

