# OpenReview forum: "Rethinking the One-shot Object Detection: Cross-Domain Object Search"
_acmmm.org/ACMMM/2024/Conference — MM2024 Poster_

### Official Review · Reviewer_4nUK · 2024-05-12

**Rating:** 4
**Confidence:** 3

**Summary:**

The paper introduces a new problem termed Cross-Domain Object Search (CDOS), which evolves the conventional One-shot Object Detection (OSOD) to accommodate object searches across varying image styles and domains, without the requirement that the target image contains objects of the same category as the query patch. The authors develop a new method involving foreground-background contrastive learning and feature augmentation to address the challenges of category and domain gaps. They also create a benchmark dataset, the Multi-Style Object Search Benchmark (MSOSB), to facilitate the training and testing of CDOS methodologies.

**Strengths:**

1. The paper introduces a new problem, CDOS, which expands the scope of one-shot object detection and aligns more closely with real-world scenarios.

2. The approach integrates foreground-background contrastive learning with domain-generalized feature augmentation, innovatively addressing the dual challenges of object category and domain disparities in object detection.

3. The paper build a new dataset namely Multi-Style Object Search Benchmark (MSOSB), which provides a benchmark to facilitate the training and testing of CDOS problems.

**Limitations:**

1. The introduction section does not clearly explain the rationale behind the proposed approach and how it enables the detection of novel object categories.

2. The evaluation section lacks comprehensive comparisons with state-of-the-art OSOD methods, especially those published after 2022.

3. The lack of code and detailed implementation details makes it difficult to reproduce the proposed method and verify the reported results.

4. Many spelling mistakes have been made throughout the manuscript, such as "pen-domain," affecting the readability and clarity.

5. It is necessary to provide details regarding how to implement stable diffusion for the generation of images.

6. Can the authors provide examples of failure cases or challenging scenarios where the proposed method struggles, and discuss potential ways to address these limitations?

**Suitability:**

1

---

### Official Review · Reviewer_fF3Q · 2024-05-31

**Rating:** 4
**Confidence:** 2

**Summary:**

This paper promotes a new task: cross-domain Object Search (CDOS). The author developed a benchmark dataset and a baseline. The proposed method is better than the compared methods.

**Strengths:**

The paper is straightforward and easy to read.
The proposed dataset is good and seems can be used for some purposes.
The proposed method achieved good performance, outperforming the compared methods on the CDOS task

**Limitations:**

The link to the proposed benchmark in the paper is broken.
The writing should be revised to correct English writing.
Compared methods are old, such as CoAE (2019), AIT (2021), BHRL (2022).

**Suitability:**

2

---

### Official Review · Reviewer_fDBn · 2024-06-01

**Rating:** 3
**Confidence:** 3

**Summary:**

The paper introduces a novel approach to One-shot Object Detection (OSOD) by proposing a new problem termed Cross-Domain Object Search (CDOS). Unlike traditional OSOD, which assumes similar image styles between the query patch and target images, CDOS relaxes these constraints by allowing different image styles and decoupling object categories between the query patch and target images. The authors present a method incorporating foreground-background contrastive learning heads and a domain-generalized feature augmentation technique to address the object category gap and domain distribution gap. Additionally, they built a new benchmark dataset, the Multi-Style Object Search Benchmark (MSOSB), to evaluate their method, showing significant performance improvements over existing methods.

**Strengths:**

•	Novelty:
The introduction of CDOS addresses practical limitations in OSOD by allowing for varied image styles and categories, which is a significant and innovative extension of the OSOD framework.

•	Technical Correctness:
The use of foreground-background contrastive learning heads enhances the network's ability to classify objects correctly and distinguish backgrounds, reducing false positives. The domain-generalized feature augmentation technique is another strong point, improving the model's robustness against unseen domains.

•	Adequate Evaluation:
The authors present a new benchmark dataset, MSOSB, which is crucial for evaluating CDOS. The extensive experiments demonstrate the effectiveness of their method, with significant performance improvements compared to state-of-the-art methods.

•	Clarity:
The paper is well-structured and clearly explains the motivation, methodology, and results. The use of figures and detailed descriptions helps in understanding the complex concepts introduced.

•	Applications:
The proposed CDOS has broad applicability in real-world scenarios where object categories and image styles are diverse, such as in search engines, surveillance, and autonomous driving.

**Limitations:**

•	Lack of Comparison with Broader Range of Methods: While the paper shows improvements over existing OSOD methods, a comparison with more diverse methods, including those from related fields like domain adaptation and generalized object detection, would strengthen the evaluation.

•	Scalability Concerns: The approach might face scalability issues when applied to extremely large datasets or real-time applications due to the complexity of contrastive learning and feature augmentation.

•	Benchmark Dataset Size and Variety: Although MSOSB is a significant contribution, its size and the variety of styles might still be limited compared to the vast diversity encountered in real-world applications. Expanding this dataset further could provide a more comprehensive evaluation.

•	Domain Generalization Challenges: The paper addresses domain gaps through feature augmentation, but the effectiveness of this approach in highly divergent domains (e.g., medical imaging vs. natural scenes) remains unclear.

**Suitability:**

3

---

### Official Review · Reviewer_J9hn · 2024-06-01

**Rating:** 5
**Confidence:** 4

**Summary:**

This paper proposes a new problem that advances over one shot object detection through what is termed cross-domain object search in which the style variation is high and the target object may or may not be in the target image. The authors propose a new method consisting of a new way to contrastively learn foreground vs background and a domain-generalized feature augmentation technique. They build a new benchmark that is better than competing methods. The authors present convincing experimental results.

**Strengths:**

1. Well written paper.
2. Sound technical approach and interesting innovations.
3. Decent advance over state of the art methods for setting up benchmark.
4. Interesting new problem.

**Limitations:**

1. While this work is interesting, with the advent of Large Vision-Language models, such work seems to be a bit behind the times. However, I am open to being convinced otherwise.

**Suitability:**

3

---

### Official Review · Reviewer_4urF · 2024-06-04

**Rating:** 4
**Confidence:** 2

**Summary:**

The paper extends one-shot object detection (OSOD) by introducing Cross-Domain Object Search (CDOS), which addresses real-world scenarios where object categories and image styles differ significantly between the query patch and target images.
The authors' method employs foreground-background contrastive learning and domain-generalized feature augmentation to bridge these gaps effectively.
They also introduce the Multi-Style Object Search Benchmark (MSOSB) to evaluate CDOS methods, demonstrating their approach's superior performance.
Key contributions include defining the CDOS problem, developing a baseline method, and creating the MSOSB dataset for comprehensive evaluation.

**Strengths:**

The introduction of the Cross-Domain Object Search (CDOS) problem is both interesting and new, addressing practical scenarios where object categories and image styles between the query patch and target images can significantly differ.
The paper is well-written and easy to read.
The Multi-Style Object Search Benchmark (MSOSB) provides a new dataset and benchmark to facilitate the training, testing, and evaluation of CDOS methods

**Limitations:**

The novelty of the proposed method remains somewhat weak as it builds upon existing techniques like contrastive learning and domain generalization without distinctly highlighting how these implementations differ substantially from prior work

**Suitability:**

2

---

### Official Review · Reviewer_s4S4 · 2024-06-10

**Rating:** 2
**Confidence:** 3

**Summary:**

- The paper is based on the one-shot object detection (OSOD) to propose a new benchmark, Cross-Domain Object Search, where the object category and the query may differ significantly (real and sketch). To solve the new problem and OSOD, the authors apply feature augmentation and contrastive learning in both foreground and background. The experiments are conducted in MS COCO and the new benchmark to demonstrate their performance.

**Strengths:**

- The new dataset seemed to be beneficial when one-shot object detection models should have the ability to detect objects belonging to a particular class in different domains.

**Limitations:**

- The paper needs to provide the implementation in detail. What is the fusion module and the value of hyper-parameter T in Equation (2)?

- Insufficient of related work. Please update the literature review with the latest related work. Some OSOD work like [32] performs the experiments on two standard datasets (PASCAL VOC, MS COCO) to demonstrate the effectiveness of their methods. The paper should conduct experiments on PASCAL VOC to provide comprehensive observations.

- Lack of ablation study. The contrastive learning for background (BG) and foreground (FG) significantly improves the model performance, and both are based on the memory bank with the key hyper-parameter being the length of the queue. These hyper-parameters for FG and BG are 128 and 65536, respectively. Does the performance of the model enhance when the length of FG increases?

**Suitability:**

1

---

### Meta-Review · Area_Chair_yNUe · 2024-07-01

**Recommendation:** Accept (Poster)
**Confidence:** 5

**Metareview:**

This paper has received one WA, three BAs, one WR and one BR as initial scores.

Pros:
The proposed dataset provides a benchmark to facilitate the training and testing of CDOS problems.
The proposed method achieved good performance.
The paper is easy to read.

Cons:
Reviewers s4S4, fDBn, and fF3Q concern about the experiments.
Reviewer 4urF highlights the lack of novelty.
Reviewers fF3Q and 4nUK find many spelling mistakes.

The authors provide a rebuttal which addresses the reviewers' concerns. The AC agrees with the majority of reviewers that the paper should be accepted.